# Preparation, Characterization, and In Vivo Evaluation of Amorphous Icaritin Nanoparticles Prepared by a Reactive Precipitation Technique

**DOI:** 10.3390/molecules26102913

**Published:** 2021-05-14

**Authors:** Cheng Tang, Kun Meng, Xiaoming Chen, Hua Yao, Junqiong Kong, Fusu Li, Haiyan Yin, Mingji Jin, Hao Liang, Qipeng Yuan

**Affiliations:** 1College of Life Science and Technology, Beijing University of Chemical Technology, Beijing 100029, China; tczy033@126.com; 2Beijing Shenogen Pharmaceutical Co., Ltd., Beijing 102206, China; kun.meng@shenogen.com (K.M.); xiaoming.chen@shenogen.com (X.C.); hua.yao@shenogen.com (H.Y.); Junqiong.kong@shenogen.com (J.K.); fusu.li@shenogen.com (F.L.); Haiyan.yin@shenogen.com (H.Y.); mingji.jin@shenogen.com (M.J.)

**Keywords:** icaritin, anti-hepatoma drug, flavonoids, amorphous nanoparticles, reactive precipitation technique

## Abstract

Icaritin is a promising anti-hepatoma drug that is currently being tested in a phase-III clinical trial. A novel combination of amorphization and nanonization was used to enhance the oral bioavailability of icaritin. Amorphous icaritin nanoparticles (AINs) were prepared by a reactive precipitation technique (RPT). Fourier transform infrared spectrometry was used to investigate the mechanism underlying the formation of amorphous nanoparticles. AINs were characterized via scanning electron microscopy, X-ray powder diffraction, and differential scanning calorimetry. Our prepared AINs were also evaluated for their dissolution rates in vitro and oral bioavailability. The resultant nanosized AINs (64 nm) were amorphous and exhibited a higher dissolution rate than that derived from a previous oil-suspension formulation. Fourier transform infrared spectroscopy (FTIR) revealed that the C=O groups from the hydrophilic chain of polymers and the OH groups from icaritin formed hydrogen bonds that inhibited AIN crystallization and aggregation. Furthermore, an oral administration assay in beagle dogs showed that C_max_ and AUC_last_ of the dried AINs formulation were 3.3-fold and 4.5-fold higher than those of the oil-suspension preparation (*p* < 0.01), respectively. Our results demonstrate that the preparation of amorphous drug nanoparticles via our RPT may be a promising technique for improving the oral bioavailability of poorly water-soluble drugs.

## 1. Introduction

Icaritin is a flavone that is derived from the Chinese herb, *Epimedium*, and exhibits important biological activities and pharmacological actions (Figure 1). As a novel anti-cancer molecule, icaritin has been shown to suppress hepatocellular carcinoma (HCC) initiation and malignant growth through the Interleukin-6/Janus-activated kinases 2/Signal transducer and activator of transcription 3 pathway [1]. In addition, icaritin has been found to ameliorate osteoporosis via estrogen-receptor-dependent and -independent pathways [2] as well as to treat erectile dysfunction by inhibiting human phosphodiesterase-5 [3].

Icaritin exhibits poor solubility and low membrane permeability (water solubility < 10 μg/mL, logP = 3.86 and pKa = 6.22) [4], as it is a flavonoid aglycone. Thus, icaritin is classified as a Biopharmaceutics Classification System (BCS)-IV compound. Currently, icaritin is in a phase-III clinical trial for the treatment of HCC in China (registered under CTR20170668 and CTR20170667), and the reported data from a phase-I/II clinical trial were promising [5]. A soft capsule prepared by a mixture of corn oil and micronized icaritin has been used in the clinical formulation. However, it showed poor oral bioavailability in beagle dogs, as well as in humans (≈2% estimated by total-plasma drug concentrations). For this reason, there is a pronounced need for a formulation with high bioavailability. As reported, icaritin nanocrystals with mean size of 220 nm were prepared as reported. However, its oral bioavailability was only enhanced 2-fold over icaritin suspension [6].

The preparation of amorphous icaritin nanoparticles (AINs) may represent a strategy for enhancing the solubility and bioavailability of icaritin. To date, various methods of preparing nanosized drugs have been reported, including supercritical fluid techniques [7,8], reactive precipitation techniques (RPTs) [9], anti-solvent precipitation methods [10,11], sonoprecipitation methods [7], evaporative precipitation methods [12], as well as media milling and high-pressure homogenization [13]. In order to further improve the solubility and dissolution rates of poorly soluble drugs, inhibition of crystallization to form amorphous particles has also been adopted widely. In practical applications, polymers are often selected as stabilizers to coat drug particles and subsequently form amorphous particles [14], or to dissolve the drug at high temperature and then cool it to form a solid dispersion [15].

In the present study, we aimed at improving the oral bioavailability of icaritin via a combination of nano- and amorphous-based effects. RPT was first developed to prepare amorphous flavonoid nanoparticles. A schematic illustration of the construction of AINs and a diagram of the experimental process are shown in Figure 2. This technique, a bottom–up method, was more amenable for preparing nanosized amorphous particles compared with that of media milling or other precipitation methods. Moreover, this technique does not require the consumption of organic solvents and is well suitable for large-scale production. We found that organic solvents were difficult to remove in other bottom–up methods, such as via anti-solvent precipitation or evaporative precipitation. In addition, based on the efficacy of our technique, we recommend that polymers be chosen rationally for preparing amorphous nanoparticles by our RPT. Furthermore, in our present study, our prepared AINs were also evaluated for their dissolution rates in vitro and oral bioavailability in beagle dogs.

## 2. Results and Discussion

### 2.1. Preparation and Characterization of AINs

Icaritin is soluble in alkaline solution as a sodium salt (pH ˃ 10) but is insoluble in neutral or acidic solutions (pH ≤ 7). In our present study, free icaritin precipitated when an alkaline solution of icaritin was added dropwise to an acidic solution during vigorous stirring (Figure 2a). This process of acid–base precipitation is a type of reactive precipitation [9]. Therefore, our RPT is suitable for icaritin. However, we found that the icaritin particles were crystals with a mean particle size of 500 nm, according to the scanning electron microscope (SEM) and X-ray powder diffraction (XRPD) results during mixing of a sodium hydroxide solution of icaritin with a hydrochloric acid solution (Figure 3). Therefore, suitable excipients, such as polymers, must be chosen to inhibit crystallization, and the proportions of icaritin and polymers should be optimized. In addition, we also compared the efficacies of other preparation methods compared to that of our RPT.

It has been reported that polymers can be absorbed on the surface of nanoparticles immediately during nanoparticle formation and can inhibit particle growth due to Ostwald ripening [16,17,18]. In addition, suppression of the crystallization of drugs results from high-energy sites on the particle surface being occupied by polymers [19]. For further reducing the particle size of icaritin and forming amorphous particles, we chose six kinds of pharmaceutical polymers (PEG4000, P407, P188, CMC-Na, soluplus, and VA64) to add into our system. In our present study, XRPD was used to analyze AINs prepared from different polymers in order to compare the differences between distinct crystal structures. As shown in Figure 4a, the XRPD patterns of AINs prepared by soluplus and VA64 both comprised diffuse peaks. These results indicated that soluplus and VA64 completely inhibited icaritin crystallization and promoted the formation of amorphous particles. Although the other four tested polymers (P188, P407, PEG4000, and CMC-Na) have been widely used to prepare amorphous particles [20], they did not work in our experiments for preparing AINs. In addition, we also measured the particle-size distributions of AINs prepared by different polymers. Only the AINs prepared with soluplus or VA64 exhibited nanosized particles (Figure 4b). The AINs composed of soluplus and icaritin exhibited the smallest particle sizes (mean size: 64 nm) with a narrow size distribution. Therefore, the AINs used for our subsequent experiments (i.e., process optimization, characterization, dissolution testing, and pharmacokinetic studies) were prepared with soluplus. Additionally, we further investigated why some polymers worked and others did not work for the preparation of amorphous nanoparticles.

Although many studies have attempted to prepare amorphous nanoparticles from polymers, there have been few studies on the mechanisms of interactions between nanoparticles and polymers. For example, the mechanisms by which polymers prevent the aggregation and crystallization of particles have only been elucidated from the perspectives of thermodynamics and kinetics [18,21]. However, the molecular mechanisms of interactions and the principles of polymer selection for preparing amorphous nanoparticles have not yet been reported.

For clarifying the mechanism of the formation of amorphous nanoparticles, we further used Fourier transform infrared spectroscopy (FTIR) to analyze all samples and compared their physical mixtures (Figure 5a). All the physical mixtures of icaritin and polymers were prepared with the same proportions of AINs. The chemical structure of icaritin has three -OH groups (3-OH, 5-OH, and 7-OH), as presented in Figure 1a. We found that the 5-OH and the neighboring C=O were attracted to one another and formed intramolecular hydrogen bonds (Figure 5b). Therefore, we found that the FTIR spectrum of pure icaritin exhibited a strong and wide -OH-stretching peak at 3311 cm^−1^ (Figure 5b). In addition, all the physical mixtures prepared by mixing icaritin and appropriate polymers revealed the same vibration at 3312–3315 cm^−1^. However, the AINs prepared with soluplus or VA64 exhibited broader and stronger bands, most obviously in the -OH-stretching region, as contrasted with those in their physical mixtures. Furthermore, the peaks of AINs at 1731–1733 cm^−1^, which belonged to the C=O stretch vibrations of soluplus and VA64, became much stronger than those of their physical mixtures. This phenomenon occurred due to hydrogen bonding between the C=O of these two surfactants and the –OH of icaritin, as shown in Figure 5b [22]. Nevertheless, there was no obvious difference in the FTIR spectra between AIN and physical mixtures prepared by CMC-Na, which indicated that hydrogen bonding was absent, although the C=O group was present in the hydrophilic chain of CMC-Na. This likely resulted from the fact that only C=O from the hydrophobic chains of polymers could access the –OH from the hydrophobic drug to then form hydrogen bonding. The C=O group of CMC-Na was from a hydrophilic carboxylate, and the C=O of soluplus and VA64 were both from hydrophobic esters. In addition, it is important to note that the results obtained from FTIR spectra were also consistent with those from the above XRPD patterns (Figure 4a). Therefore, our findings indicate two principles for rationally choosing polymers to prepare amorphous nanoparticles for poorly water-soluble drugs. First, the polymer must contain long hydrophobic chains to coat the hydrophobic drug and inhibit the aggregation of molecules. Second, if there are hydrogen-bond acceptors or donors in the chemical structure of the drug in question, it is better to choose polymers with hydrogen-bond donors or acceptors in the hydrophobic chain. The hydrogen-bonding force between a polymer and drug is much stronger than the hydrophobic force between them, which may further prevent drug molecules from undergoing orderly aggregation and crystallization.

In addition to our RPT, we also tested three other methods (media milling, antisolvent precipitation, and sonoprecipitation) for preparing amorphous nanoparticles. Media milling was first used to prepare AINs. The size of nanoparticles obtained by media milling was approximately 120 nm (Figure 6b), but these nanoparticles were still crystals according to our XRPD results (Figure 6a). Media milling is a top–down method [16]. During this process, large crystals are broken into smaller crystals via high energy. However, it is difficult to transform drugs that crystalize easily into to an amorphous form. For this reason, we were unable to obtain AINs by media milling alone. In addition to this top–down method, we also tested two bottom–up methods (i.e., anti-solvent precipitation and sonoprecipitation). Using these bottom–up methods, particles were produced at the molecular level, and polymers were used to coat and stabilize the particles, which have been suggested to represent promising methods for preparing AINs [16]. However, our results revealed that AINs prepared by anti-solvent precipitation and sonoprecipitation methods yielded obvious crystals, as shown in Figure 6c. These findings are consistent with those of Wu et al. [20], who prepared apigenin (a flavonoid compound) nanoparticles by anti-solvent precipitation and found that the resultant morphology was also in the form of crystals. Thus, our results indicate that these other commonly used methods did not work for preparing AINs.

Based on the results of the above experiments, we conclude that our RPT is the alternative method for preparing AINs. In this process, a high degree of supersaturation was generated immediately due to the decreased solubility of icaritin by acid–base neutralization reactions, which results in a fast nucleation rate and ultrafine particles coated with polymers [9]. Moreover, toxic organic solvents were avoided in the preparation. However, in the anti-solvent precipitation and sonoprecipitation process, organic solvents are commonly used to dissolve poorly soluble drugs [11,20]. Therefore, our RPT represents the most promising method for preparing amorphous nanoparticles from water-insoluble drugs such as icaritin.

According to the results of Figure 4, soluplus was selected as the optimized polymer to prepare AINs. For further obtaining optimal amorphous particles and to improve the drug-loading of AINs, the amount of soluplus was optimized. When the volume of soluplus solution (acidic solution) was maintained and the concentration of soluplus was increased from 0.3% to 0.8%, AINs changed markedly from a crystal form to an amorphous state, according to our XRPD results (Figure 7a). Thus, a 0.8% soluplus aqueous solution was the most suitable concentration to provide enough steric hindrance for preventing particle growth and the occurrence of aggregation. Higher concentrations of soluplus decreased drug-loading, although optimal amorphous icaritin was also yielded from a 0.8% soluplus aqueous solution. Figure 7b shows XRPD patterns of AINs prepared at different volumes of soluplus aqueous solutions when the concentration of soluplus was sustained. The crystallinity decreased as a function of the volume of soluplus solution. The optimal volume of soluplus aqueous solution was 60 mL, whereas higher volumes of soluplus decreased drug-loading.

Via our optimized process, lyophilized AINs were then prepared. The icaritin content, equivalent to drug-loading, was 31% (as quantified by High-Performance Liquid Chromatography (HPLC)). The lyophilized AINs were drug-rich particles that were similar to those via amorphous solid dispersion (ASD) prepared by hot-melt extrusion [23].

Drug degradation is a legitimate concern because strong acids or bases can destroy drug particles, which restricts the use of RPTs in the pharmaceutical industry. Figure 8a shows the HPLC photographs of lyophilized AINs and raw icaritin. The purity of lyophilized AINs was 99.5%, which was close to that of raw icaritin (99.6%). Some impurities increased weakly, but each single impurity’s content was no more than 0.20%. Therefore, the degradation of icaritin resulting from the acid–base reaction process was acceptable according to International Council for Harmonisation of Technical Requirements for Pharmaceuticals for Human Use (ICH) Q3B. These results suggest that bioactive flavonoids may tolerate acid–base reactions and that our RPT is an alternative approach for such compounds.

The crystalline structures of raw icaritin and lyophilized AINs were analyzed by comparing their XRPD and differential scanning calorimetry (DSC) profiles. Figure 8b shows the XRPD patterns of raw icaritin, lyophilized AINs, and the physical mixtures of raw icaritin and soluplus. The raw icaritin and the physical mixtures exhibited sharp crystalline peaks between 4° and 40°. However, only a diffuse peak was detected in the pattern of the lyophilized AINs, which demonstrated that the prepared lyophilized AINs were in the desired amorphous form. This finding was also confirmed by comparing the DSC thermal profiles of raw icaritin and lyophilized AINs (Figure 8c). There was an endothermic band around 254.70 °C for raw icaritin, which indicated that raw icaritin was in the crystalline form. In contrast, no endothermic band was present for lyophilized AINs, confirming that the lyophilized AINs were amorphous.

SEM images of AIN suspensions are presented in Figure 8d. Raw icaritin was in the form of micronized square crystals that were regularly ordered and had high density. This crystal structure led to the slow dissolution and low solubility of icaritin. However, AINs were homogeneous and spheroidal nanoparticles (40–50 nm), as presented in Figure 8d. Thus, AINs prepared by our optimized process were amorphous nanoparticles.

### 2.2. In Vitro Dissolution Testing

Drug-dissolution testing in vitro is a prerequisite for the use of solid oral forms of drugs and is routinely used during all phases of development for formulation research and development. At early stages of development, appropriate conditions in vitro that emulate conditions in vivo are selected to predict drug-release profiles in vivo.

A sink condition was obtained by addition of 0.3% (v/v) dodecayl dimethyl amine oxide (DDAO) in simulated intestinal fluid (pH 6.8, PBS solution). Lyophilized AINs and icaritin oil suspension were dissolved in this solution as free icaritin rather than encapsulated nanoparticles. The dissolution rates of lyophilized AINs and an oil suspension of icaritin from capsules were 82.2 ± 1.1% and 12.4 ± 2.9%, respectively, in the initial 60 min (Figure 9). The dissolution rate of lyophilized AINs was significantly faster than that of the oil suspension, the latter of which was composed of icaritin microcrystals and corn oil.

The amorphous state is a metastable state that allows for high supersaturation in contrast to that of the crystal form [17]. This increased supersaturation in the gastrointestinal tract will lead to a higher concentration gradient and higher bioavailability [14]. Compared with those of larger-sized particles, nanosized particles provide greatly enlarged surface areas and produce rapid dissolution rates [24]. Our results showed that the surfactants on the surface of nanoparticles exhibited hydrophilic properties resulting from polyethylene glycol (PEG) block of soluplus, and these particles were prone to dispersion in water. Nevertheless, the oil suspension showed hydrophobic properties and was easy to aggregate and adhere to the rotating baskets.

### 2.3. In Vivo Pharmacokinetic Studies

In order to investigate whether lyophilized AINs can improve oral bioavailability, the pharmacokinetics in beagle dogs was studied because the pharmacokinetic parameters in beagle dogs are similar to those in humans, as compared with those in mice or rats (unpublished data). Six beagle dogs were involved in the present study and were reused once after a washing-out period (one week). The pharmacokinetic parameters and comparisons are listed in Table 1. Curves of the average plasma concentrations over time are shown in Figure 10. After oral administration, C_max_ of lyophilized AINs in the fasted state was 209 ± 109 ng/mL, which was 3.3-fold higher than that of the oil suspension (*p* < 0.01, Table 1). AUC_last_ of lyophilized AINs was 1279 ± 739 h·ng/mL, which was 4.5-fold higher than that of the oil suspension (*p* < 0.01, Table 1). These results confirmed that lyophilized AINs significantly enhanced the adsorption of icaritin in vivo, which was consistent with the results of our dissolution testing in vitro. This increased bioavailability was due to the faster dissolution rate and greater supersaturation of icaritin from lyophilized AINs in the gastrointestinal tract.

Icaritin, as a BCS-IV drug, has three primary limiting steps (poor solubility, poor permeability, and P-gp substrate) for absorption. In terms of poor solubility as a limiting step, good solubility in the gastrointestinal tract is a pre-condition to high bioavailability for any drug, but the solubility of icaritin is less than 10 μg/mL in water (pH ≤ 7). AINs significantly improved the solubility of icaritin, as revealed by our dissolution testing in vitro. In contrast, the oil suspension of icaritin presented a low bioavailability due to its poor solubility. In terms of poor permeability as a limiting step, icaritin shows low membrane permeability (*P*_app (A→B)_ ≈ 10^−8^ cm/s) because extremely hydrophobic icaritin is unable to sufficiently access the unstirred water layer of the intestinal membrane [25]. However, AINs coated with hydrophilic polymers contribute to transport across hydration layers. As reported, the amorphous nanoparticles of cefuroxime and atorvastatin calcium were prepared without polymers by Dhumal et al. [7] and Kim et al. [26], and the bioavailabilities of amorphous nanoparticles of cefuroxime and atorvastatin calcium were both enhanced just one time. Therefore, optimal polymers used in amorphous nanoparticles are essential for drugs with poor permeabilities. In terms of the P-gp substrate as a limiting step, icaritin is a P-gp substrate (as are other flavonoid compounds) that is pumped out of intestinal cells even if it is initially absorbed within these cells. Therefore, the improved solubility of icaritin will lead to a higher drug concentration in the small intestine, which will bind to and saturate efflux transporters. In addition, the polymers themselves from amorphous nanoparticles, including soluplus, have been reported to be P-gp inhibitors that inhibit P-gp efflux functionality via interacting with cell membranes to alter the membrane microenvironment [27,28].

Thus, the preparation of drug-rich amorphous nanoparticles with polymers is a promising approach for improving the bioavailability of water-insoluble drugs. High bioavailability has been found to contribute to higher blood concentration of these water-insoluble drugs and so to better efficacy. In addition, these nanoparticles could be directly transported across intestinal epithelium through transcytosis, which is not possible with the traditional formulation [29]. The absorbed nanoparticles had the advantages of passive tumor targeting and long circulation because of the enhanced permeability and retention (EPR) effect [30].

## 3. Materials and Methods

### 3.1. Materials

Icaritin (99% purity, micronized crystals) was provided by Beijing Shenogen Pharmaceutical Co. Ltd. (Beijing, China). Soluplus, poloxamer 188 (P188), poloxamer 407 (P407), VA64, PVP K90, sodium laurylsulfonate (SLS), and carboxymethyl cellulose sodium (CMC-Na) were purchased from Beijing Fengli Jingqiu Pharmaceutical Co., Ltd. PEG4000 was purchased from Nanjing Weier Pharmaceutical Co., Ltd. DDAO was purchased from Maya Reagent Co. (Beijing, China). β-glucuronidase (Type H-2) was purchased from Sigma Chemical Co. (St. Louis, MO, USA). HPLC solvents were purchased from Fisher Scientific. A SunFire C18 column (4.6 × 250 mm, 5 μm) and an Eclipse Plus C18 column (4.6 × 100 mm, 3.5 μm) were purchased from Waters Technologies Inc. (Milford, MA, USA) and Agilent Technologies, respectively.

### 3.2. Preparation of AINs

#### 3.2.1. Reactive Precipitation Technique (RPT)

Icaritin (300 mg) and NaOH (120 mg) were dissolved in 15 mL of water (base solution). Excipient (480 mg) and concentrated hydrochloric acid (250 μL) were dissolved in 60 mL of water (acidic solution). This basic solution with icaritin was added to the aforementioned acidic solution via a peristaltic pump under rapid stirring (1000 rpm). After approximately 10 min of stirring, the suspension was lyophilized for 24 h. The experimental process for the preparation of AINs is illustrated in Figure 2.

For obtaining optimal amorphous nanoparticles and improving drug-loading, the amount of soluplus that was used was optimized. The concentration of soluplus in acidic solution was first adjusted to 0.3%, 0.8%, 1.2%, and 1.6%, while the volume of the acidic solution was maintained. Subsequently, the volume of acidic solution was adjusted to 15, 30, 60, 75, and 90 mL with the optimized concentration of soluplus. All other procedures were the same as those described above.

#### 3.2.2. Media Milling

CMC-Na (2.0 g), SLS (1.0 g), and PVP K90 (0.17 g) were dissolved in 200 mL of water. Then, icaritin (6 g) was added to the solution and stirred for 30 min. Then, the resulting homogenous suspension was milled by a Dyno-mill (WAB Group, Muttenz, Switzerland) at 4000 rpm for 30 min.

#### 3.2.3. Anti-Solvent Precipitation

Icaritin (300 mg) was dissolved in 3 mL of tetrahydrofuran and was then added dropwise to 50 mL of 0.8% soluplus aqueous solution over the course of 15 min during vigorous stirring (1000 rpm). The obtained suspension was evaporated overnight at room temperature and lyophilized for 24 h.

#### 3.2.4. Sonoprecipitation

Icaritin (300 mg) was dissolved in 3 mL of tetrahydrofuran and was then added dropwise to 75 mL of 0.8% soluplus aqueous solution over the course of 10 min in a sonoreactor (40 kHz, 500 W). The suspension was evaporated overnight at room temperature and then lyophilized for 24 h.

### 3.3. Characterization of AINs

In our study, the particle size distribution (PSD), morphology, crystalline state, drug/polymer interactions, contents, and impurities of AINs were investigated.

The average size of nanoparticles in suspension was measured by dynamic light scattering with a Nano ZS (Malvern Instruments, Malvern Worcestershire, UK) at a wavelength of 633 nm at 25 °C. Prior to measurement, samples were diluted with purified water. All measurements were repeated three times, and the average value was used.

The surface morphologies of AINs were observed using a TESCAN MAIA3 field-emission scanning electron microscope (SEM, TESCAN, Brno, Czech Republic). Prior to observation, samples were coated with a layer of gold using a precision-etching coating system (LEICA EM SCD 500, Leica Microsystems, Wetzlar, Germany). Surface morphologies were obtained at 15 kV.

XRPD was utilized to determine the crystallinity of the lyophilized powders. X-ray diffraction patterns were obtained using an X-ray diffractometer (Ultima III, Rigaku, Tokyo, Japan) with Cu-Ka radiation, a voltage of 40 kV, and a current of 40 mA. All scans were performed with a scanning rate of 4°/min with steps of 0.02° from 3° to 40° at 2*θ* ranges.

DSC curves of the samples were recorded by a thermal analysis system (Pyris 6, Perkin-Elmer, Waltham, MA, USA). The samples were placed in an aluminum pan and heated at a rate of 10 °C/min between 30 and 400 °C under a nitrogen atmosphere, which was maintained by nitrogen gas (25 mL/min). Instrument calibrations with respect to enthalpy and temperature were achieved by high-purity indium.

Drug/polymer interactions were recorded via FTIR (AVATAR 370 DTGS, Thermo Nicolet, Waltham, MA, USA) in a range of 400–4000 cm^−1^ at a resolution of 4 cm^−1^. All samples were diluted 100-fold in KBr powder. FTIR was obtained in a KBr disc.

The content and impurity of icaritin were determined by an HPLC method (Agilent 1260 series, Agilent, Santa Clara, CA, USA). A SunFire C18 column (4.6 × 250 mm, 5 μm, Waters, Milford, MA, USA) was used for chromatographic separation. The mobile phase consisted of solvent A (0.1% phosphoric acid solution) and solvent B (methanol: tetrahydrofuran = 35:26, containing 0.1% phosphoric acid). A gradient program was used for the HPLC separation at a flow rate of 1.2 mL/min. The mobile phase was initially composed of 50% B and was then linearly increased to 62% B within 10 min and maintained for 45 min, after which it was reduced to 50% B within 0.1 min and was then stopped after 55 min. The ultraviolet detector was set at 373 nm. The injection volume was 20 μL, and the column temperature was 35 °C. The sample was dissolved with methanol. The standard calibration curve was built within concentrations of 2.1–343.1 μg/mL. The typical regression equation for icaritin was Y = 48,514.25X − 44.38 (*R*^2^ = 0.999, in which X is the concentration of icaritin and Y the peak area). The limit of detection (LOD) was 0.10 μg/mL and the limit of quantitation (LOQ) was 0.25 μg/mL.

### 3.4. In Vitro Dissolution Testing

Dissolution testing was carried out using a USP Apparatus I (Baskets) method (RC806D, Tianjin, China) to investigate the release of free icaritin in vitro. The basket-stirring speed and bath temperature were set at 100 rpm and 37 °C, respectively. Specifically, 500 mL of buffer (pH = 6.8) with 0.3% DDAO (w/v) was used as the dissolution medium. The dissolution samples were analyzed via HPLC (1260 series, Agilent, USA) for 0, 5, 10, 20, 30, 45, 60, 90, and 120 min.

### 3.5. In Vivo Pharmacokinetic Studies

All of animal experiments were carried out with the approval of Institutional Animal Care and Use Committee of Pharmaron Lab Animal Research in an AAALACi-accredited facility (Certification number: 001760). Male beagle dogs, weighing approximately 10 kg each, were purchased from Beijing Marshall Biotechnology Co. Ltd. Dogs were housed in a room with a controlled temperature (18–26 °C) and humidity (40–70%), were exposed to a controlled 12-h light/dark cycle, and were provided food and water ad libitum. Dogs were fasted overnight before the administration of the drug and were fed at approximately 2 h after dosing. Six dogs were first orally administrated an oil suspension of icaritin at a dose of 20 mg/kg. After the washing-out period (one week), these six dogs were then orally administrated AINs at a dose of 20 mg/kg. Blood samples (1 mL) were collected via venipuncturing of peripheral veins at 0, 0.167, 0.5, 1, 2, 4, 6, 8, 12, and 24 h after drug administrations. Then, these blood samples were centrifuged at 2000× *g* for 10 min at 2–8 °C to obtain plasma. Then, plasma samples were transferred to cryogenic vials and stored in a freezer at −75 ± 15 °C prior to their analyses.

The concentration of icaritin in plasma was determined after hydrolysis with β-glucuronidase. Specifically, 55 μL of plasma was incubated for 1 h with 10 μL of acetic acid and 25 μL of phosphate buffer (pH = 5.0) containing β-glucuronidase at 37 °C. After incubation, 200 μL of internal standard solution containing 50 ng/mL of dexamethasone was added and vortex-mixed for 2 min. The mixture was centrifuged at 4000× *g* for 15 min, and 5 μL of supernatant was injected into the LC/MS/MS system for analysis.

The LC/MS/MS analytical method that we performed was as follows. The HPLC system consisted of an LC-30D binary-pump-coupled Shimadzu SIL 30AC system (Shimadzu, Kyoto, Japan) with an API 5500 LC/MS/MS instrument (Agilent, Palo Alto, CA, USA). The column used for the separation was a Kinetex 5-μm C18 100 A (50 × 2.1 mm, Phenomenex, Torrance, CA, USA). The column temperature was maintained at 40 °C. The HPLC mobile phases consisted of solvent A (5% ACN in 0.1% formic acid) and solvent B (95% ACN in 0.1% formic acid). A gradient program was used for the HPLC separation at a flow rate of 0.6 mL/min. The mobile phase was initially composed of 20% B for 0.2 min and was then linearly increased to 100% B within 1.9 min and was maintained for 2.1 min, after which it was reduced to 20% B within 2.11 min and was stopped at 2.5 min. The total run time was 2.5 min. The eluent from HPLC was directly introduced into an electrospray–ionization interface. Nitrogen was used as the nebulizing gas at 50 psi, and the nebulizer temperature was set to 400 °C. ESI was performed in the negative mode with an ion-spray voltage of −4500 V. Multiple-reaction monitoring detection was employed using nitrogen as the collision energy, with a dwell time of 50 ms for each transition. The precursor–product ion pairs of icaritin and dexamethasone (as the internal standard substance) were m/z 367.12–297.00 and m/z 391.14–361.20, respectively. The calibration curve, Y = 0.0343X − 0.00945 (in which X is the concentration of icaritin and Y the peak area), was found to be linear in the range of 1–1000 ng/mL (*R*^2^ = 0.994). Its coefficient of variation was less than 5.5% and the accuracy was in the range of 87–107%. The method was accurate and precise. The LOQ was 1 ng/mL.

### 3.6. Pharmacokinetic Parameters and Statistical Analysis

Pharmacokinetic parameters were analyzed using DAS 2.0 software (Drug and Statistics). Two-tailed Student’s *t* tests were employed to perform statistical analyses. Differences between two groups were considered significantly different at *p* < 0.05.

## 4. Conclusions

Our present study provides the first demonstration of AINs being successfully prepared by an RPT. Six kinds of polymers were chosen to prepare AINs. The mechanisms of interactions among drugs and polymers were investigated by FTIR. The hydrogen-bonding forces between polymers and icaritin were determined and were found to underlie why AINs prepared with soluplus and VA64 were amorphous whereas those prepared from other polymers remained as crystals. Our results suggest that polymers must be chosen rationally when being used to prepare amorphous nanoparticles for poorly water-soluble drugs. In addition, our RPT was optimized and compared with the efficacies of other processes, and the results showed that our RPT was the most promising method for preparing amorphous nanoparticles for drugs with pH-dependent solubility. The physiochemical characteristics of AINs—such as their crystalline form, surface morphology, particle size, contents, and impurities—were also determined. In vitro dissolution testing revealed that AINs presented a high dissolution rate in comparison with that of an oil suspension. After oral administration in beagle dogs (*n* = 6), C_maxt_ and AUC_last_ of AINs were both significantly enhanced compared to those of the clinical formulation (*p* < 0.01), and the oral bioavailability of this new formulation was improved by 450% in comparison with that of the clinical formulation. In summary, our results demonstrate that selecting a suitable polymer and preparing amorphous drug nanoparticles via our RPT represent a promising technique for enhancing the oral bioavailability of both BCS-II and -IV drugs, especially for drugs with pH-dependent solubilities.

## Figures and Tables

**Figure 1 molecules-26-02913-f001:**
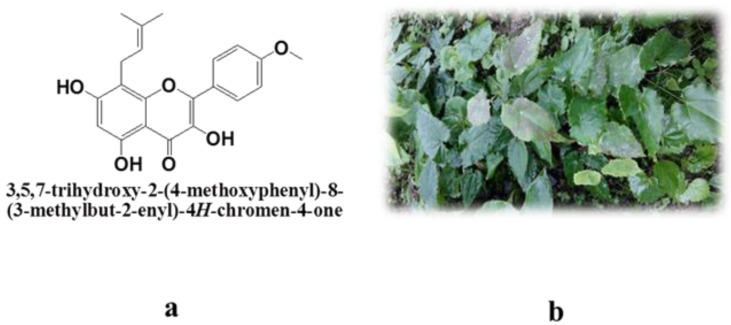
Chemical structure of icaritin (**a**) and an image of *Epimedium* (**b**).

**Figure 2 molecules-26-02913-f002:**
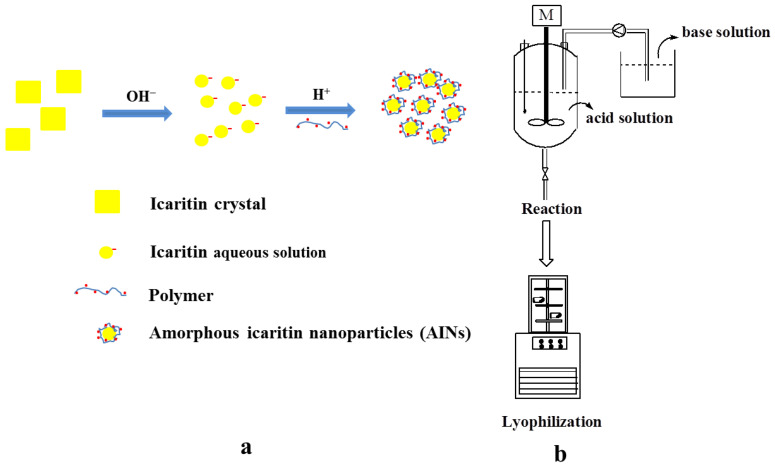
Schematic illustration of the construction of AINs (**a**) and a diagram of the experimental process for our RPT (**b**). AINs: amorphous icaritin nanoparticle; RPT: reactive precipitation technique.

**Figure 3 molecules-26-02913-f003:**
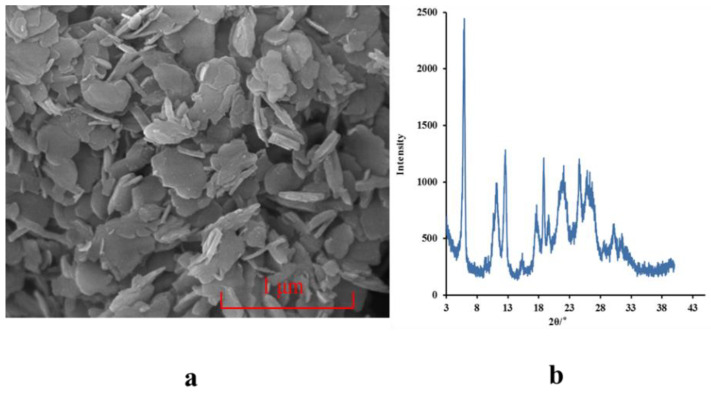
SEM image (**a**) and XRPD pattern (**b**) of icaritin particles prepared without polymers via our RPT. RPT: reactive precipitation technique; SEM: scanning electron microscope; XRPD: X-ray powder diffraction.

**Figure 4 molecules-26-02913-f004:**
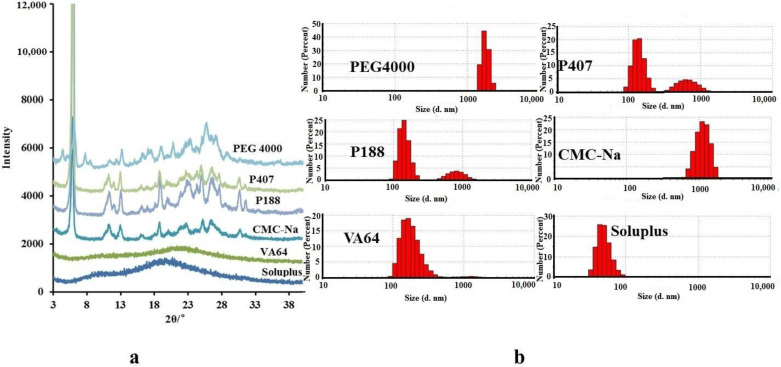
XRPD patterns (**a**) and PSD spectra (**b**) of AINs prepared with different polymers. AINs: amorphous icaritin nanoparticle; PSD: particle size distribution; XRPD: X-ray powder diffraction.

**Figure 5 molecules-26-02913-f005:**
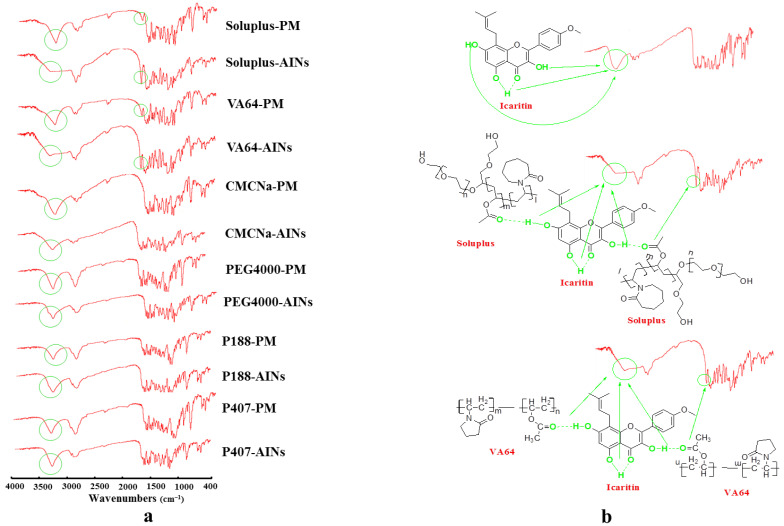
FTIR spectra of AINs and physical mixtures (PM) of icaritin and polymers (**a**), and the mechanisms of interaction between icaritin and polymers (**b**). AINs: amorphous icaritin nanoparticle; FTIR: Fourier transform infrared spectroscopy.

**Figure 6 molecules-26-02913-f006:**
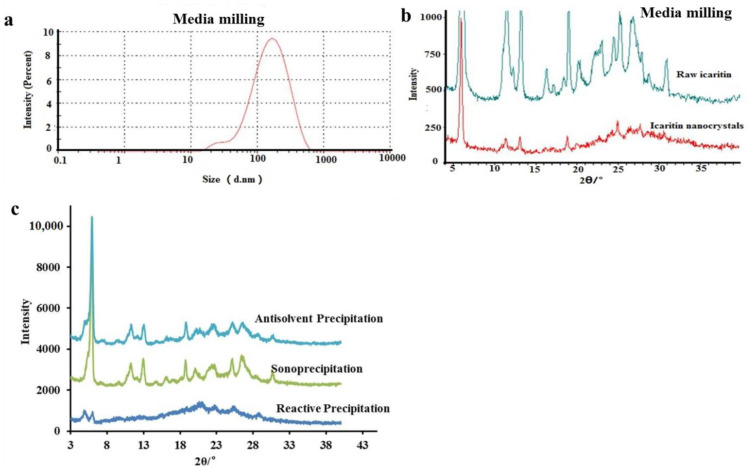
PSD and XRPD patterns of icaritin nanocrystals prepared by media milling (**a**,**b**), and XRPD patterns of AINs prepared using three other preparation methods (**c**). AINs: amorphous icaritin nanoparticles; PSD: particle size distribution; XRPD: X-ray powder diffraction.

**Figure 7 molecules-26-02913-f007:**
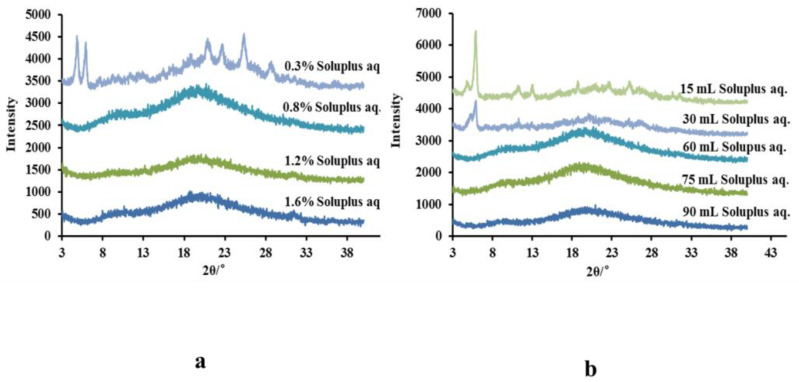
XRPD patterns of AINs prepared with different volumes of soluplus aqueous solutions (**a**) and different soluplus concentrations (**b**). AINs: amorphous icaritin nanoparticles; XRPD: X-ray powder diffraction.

**Figure 8 molecules-26-02913-f008:**
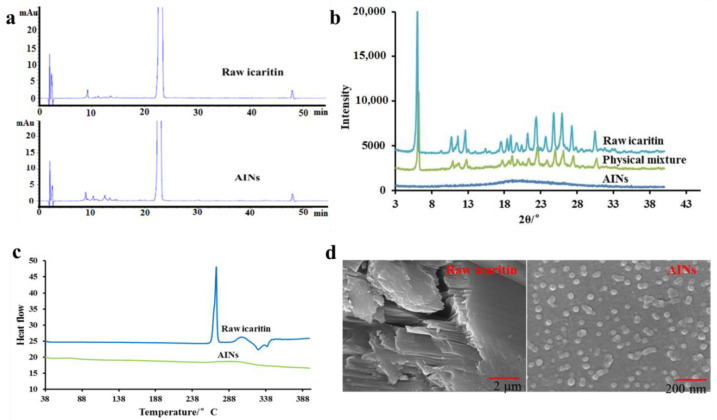
Impurity profiles of raw icaritin and AINs (**a**); XRPD of raw icaritin, physical mixture and AINs (**b**); DSC of raw icaritin and AINs (**c**); SEM of raw icaritin and AINs (**d**). AINs: amorphous icaritin nanoparticle; XRPD: X-ray powder diffraction; DSC: differential scanning calorimetry; SEM: scanning electron microscope.

**Figure 9 molecules-26-02913-f009:**
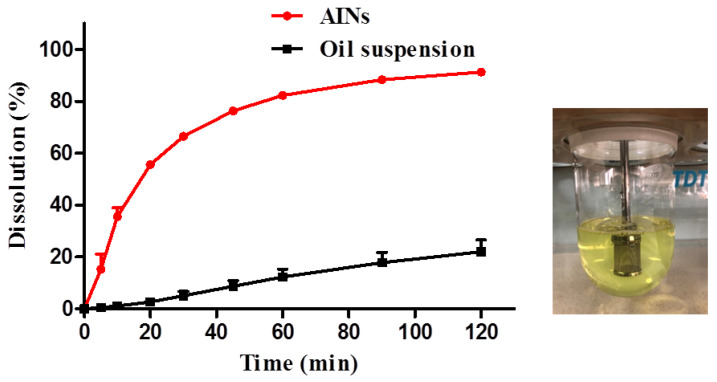
Dissolution profiles of lyophilized AINs and an oil suspension of icaritin (*n* = 6). AINs: amorphous icaritin nanoparticles.

**Figure 10 molecules-26-02913-f010:**
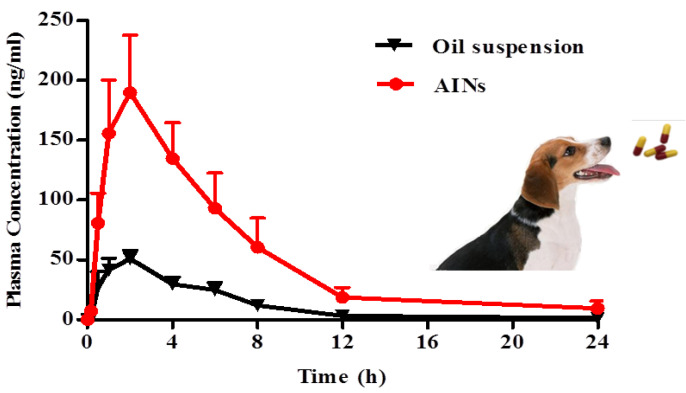
Profiles of mean plasma concentrations of icaritin versus time in beagle dogs after oral administration of AINs or an oil suspension at a dose of 20 mg/kg (*n* = 6). AINs: amorphous icaritin nanoparticles.

**Table 1 molecules-26-02913-t001:** Pharmacokinetic parameters after oral administration of AINs and an oil suspension in beagle dogs (*n* = 6).

PK Parameters	AINs	Oil Suspension
T_max_ (h)	2.50 ± 2.07	1.42 ± 0.66
C_max_ (ng/mL)	209 ± 109	63 ± 14
Terminal T_1/2_ (h)	3.80 ± 2.02	2.04 ± 0.32
AUC_last_ (h·ng/mL)	1279 ± 739	283 ± 90
AUC_INF_ (h·ng/mL)	1252 ± 952	293 ± 105

AINs: amorphous icaritin nanoparticles; AUC: area under the curve.

## Data Availability

The data presented in this study are available on request from the corresponding authors.

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
