# Peer review of "Preparation, Characterization, and In Vivo Evaluation of Amorphous Icaritin Nanoparticles Prepared by a Reactive Precipitation Technique"

_molecules, 2021, doi:10.3390/molecules26102913_

Round 1

Reviewer 1 Report

26.04.2021.

A review to evaluate its suitability for publication Type of manuscript:

Article
Title: Preparation, Characterization, and In Vivo Evaluation of Amorphous
Icaritin Nanoparticles Prepared by a Reactive Precipitation Technique

 Authors: Cheng Tang, Kun Meng, Xiaoming Chen, Hua Yao, Junqiong Kong, Fusu

Li, Haiyan Yin, Mingji Jin, Hao Liang *, Qipeng Yuan *

The section: Medicinal Chemistry

This Manuscript presents scientifically interesting results on the study of the flavanoid Icaritin (Epimedium) for the development of an oral dosage form of antitumor action with improved pharmacokinetic characteristics. The relevance of this study is beyond doubt, since the analysis of publication activity on this topic - the flavanoid icaritin - showed more than 120 scientific articles over the past 5 years (according to Pub Med.gov.)

The methodological approaches proposed by the authors are well thought out - from particle morphology to solubility and the study of bioavailability on the example of animals.

The results are consistent with the set objectives and with the main goal of improving the oral bioavailability of the poorly water-soluble drug.

However, I have a few comments that need to be included in the study.

  1. Line 41-42. In the caption to the figure - the chemical structure of icaritin, indicate its Systematic Names
  2. Line 43-44 - links to data on solubility and permeability are needed, since data differ - Water Solubility 0.00821 mg / mL (https://go.drugbank.com/drugs/DB12672). Add data by log P, pKa
  3. Line 220-221 – What is the internal standard of the HPLC method
  4. Line 255-256 In Vitro Dissolution Testing should be tested in the pH range from 1.2 to 6.8, simulating various gastrointestinal environment? Why are the results shown at only one pH value?

Regards, reviewer

Author Response

Please see the attachment, Thanks!

Reviewer 2 Report

Recommendation: The manuscript should be publishable with minor revisions.

Comments:

This paper demonstrated that the nanosized amorphous icaritin nanoparticles (AINs), prepared by a reactive precipitation technique (RPT), exhibited a higher dissolution rate and increased oral bioavailability than the traditional oil-suspension preparation. The authors first tried various pharmaceutical polymers/different methods to prepare AINs, and identified that by using their PRT, the soluplus formed the most desired amorphous/nanosized particles. After optimizing the concentration and volume of soluplus, they then prepared lyophilized AINs with good drug-loading and stability. They further validated that the AINs showed improved water dispersion in vitro and oral bioavailability in vivo. Importantly, the authors tried to clarify the mechanism of the formation of amorphous nanoparticles, which is valuable to this area. This is a well written manuscript with convincing supporting data.     

However, there are two key points should be addressed:

  1. The authors claimed that “our RPT was the optimal choice for preparing AINs” in line 189, but they did not emphasize the critical differences between theirs and the conventional RPT. They should either point out the key differences or use more compromised words.
  2. The authors should add a paragraph to discuss the potential differences of the AINs regarding the drug efficacy and biodistribution, compared to the clinical oil-formulation.

Author Response

  1. The authors claimed that “our RPT was the optimal choice for preparing AINs” in line 189, but they did not emphasize the critical differences between theirs and the conventional RPT. They should either point out the key differences or use more compromised words.

    Thank you. Your advice is very good. I did not emphasize the critical differences between our RPT and the conventional RPT. I have changed “alternative method” to “optimal choice.”

  1. The authors should add a paragraph to discuss the potential differences of the AINs regarding the drug efficacy and biodistribution, compared to the clinical oil-formulation

Thank you for these valuable contributions. I added this paragraph: “Thus, the preparation of drug-rich amorphous nanoparticles with polymers is a promising approach for improving the bioavailability of water-insoluble drugs. High bioavailability has been found to contribute to higher blood concentration of these water-insoluble drugs and so to better efficacy. In addition, these nanoparticles could be directly transported across intestinal epithelium through transcytosis, which is not possible with the traditional formulation [1]. The absorbed nanoparticles had the advantages of passive tumor targeting and long circulation because of the enhanced permeability and retention (EPR) effect [2]”.

[1] Qu, X.; Zou, Y.; He, C.; Zhou, Y.H.; Jin, Y.; Deng, Y.; Wang, Z.; Li, X.; Zhou, Y.X.; Liu, Y. Improved intestinal absorption of paclitaxel by mixed micelles self-assembled from vitamin E succinate-based amphiphilic polymers and their transcellular transport mechanism and intracellular trafficking routes. Drug Deliv. 2018, 25, 210–225, doi:10.1080/10717544.2017.1419513.

[2] Gaucher, G.V.; Dufresne, M.H.; Sant, V.P.; Kang N.; Maysinger D.; Leroux, J.C.. Block copolymer micelles: preparation, characterization and application in drug delivery. J. Control. Release, 2005, 109: 169–188, doi:10.1063/1.364067.